# Professional soccer practitioners' perceptions of using performance analysis technology to monitor technical and tactical player characteristics within an academy environment: A category 1 club case study

**Tia-Kate Davidson**[1]*, **Steve Barrett**[2], **John Toner**[1], **Chris Towlson**[1]

**1** University of Hull, School of Sport, Exercise and Rehabilitation Sciences, Hull, United Kingdom, **2** Sport Science, Performance Analysis, Research and Coaching (SPARC), Playermaker, London, United Kingdom

* t.davidson-2021@hull.ac.uk

## Abstract

This study aimed to identify professional soccer practitioners' perceptions of the application of performance analysis technology within a single academy club. Secondary aims were to understand the importance that practitioners place on monitoring technical and tactical player characteristics, current practices, and barriers to implementing wearable technology. Utilising a mixed method design, forty-four professional soccer academy practitioners (Age = 32 ± 5.8; Years of experience = 8.5 ± 6.2) completed an online survey intended to examine present trends, professional practices, and perceptions regarding the monitoring of technical and tactical metrics. Frequency and percentages of responses for individual items were calculated. Subsequently, eleven participants who were directly involved with the monitoring of players were recruited to participate in a semi-structured interview. Interview data was transcribed and analysed using a combination of deductive and inductive approaches to identify key themes. The main findings across both phases of the study were that (1) technical and tactical metrics are monitored more frequently in matches (Technical: 89%; tactical: 91%) than training (Technical: 80%; Tactical 64%), predominantly due to time constraints and staffing numbers. Accordingly, practitioners believe that it would be beneficial to have an automated way of tracking technical (79%) and tactical (71%) metrics and would consider using a foot-mounted IMU to do so (technical (68%) and tactical (57%)). (2) Monitoring technical and tactical metrics is beneficial to assist with player development and to enrich feedback provision (3) Key stake holders, coaches and players should be informed of the relevance and rationale for monitoring. (4) For successful implementation and continued uptake, the information delivered needs to be both meaningful and easy to understand. Findings suggest that although participants appreciate the need to collect technical and tactical metrics, they are keen to ensure that wearable-derived data does not replace experiential and contextual knowledge. Accordingly, practitioners need to work closely with coaches to determine the contexts in which metrics may or may not prove useful. However, as the sample comprised of participants from a single academy, further studies including more

**Data Availability Statement:** Here is the open access link for the underlying data set: https://hull-repository.worktribe.com/output/4423771.

**Funding:** The author(s) received no specific funding for this work.

**Competing interests:** The authors have declared that no competing interests exist.

practitioners are warranted. Likewise, future research could also extend to include academy soccer players perceptions too.

## Introduction

Within academy soccer exceptional individuals that are recognised as having potential for future long-term success are enrolled within specialised developmental programmes [1] such as the Elite Player Performance Plan (EPPP) [2]. The EPPP is a long-term athletic and talent development framework with the primary objective to enhance domestic professional soccer clubs to develop more and better-quality homegrown players for first team selection. Soccer clubs commonly evaluate players' technical and tactical characteristics subjectively, subsequently making selections/deselections based on their perceived capability and performance [3]. However, viewing these characteristics in isolation can lead to biased results, misjudgements, and repeated errors [4–6]. Attempts at identifying talent are traditionally informed by coaches and scouts' intuition, and gut feelings based on previous experience [7]. This decidedly subjective approach is bounded by the interpretation of actions observed that can last for split seconds [8]. To combat these limitations, it is suggested that decisions regarding players should be based on a combination of subjective evaluations and objective measures such as performance analysis and monitoring technologies [4].

Technical skills have been shown to better distinguish between competitive standards when compared to physical skills [9]. Furthermore, practitioners responsible for player development have identified technical and tactical ability as the greatest predictor of future success amongst youth soccer players [3]. Despite this, many quantitative development protocols implemented within soccer often focus on general athletic attributes (e.g., endurance, strength, speed, and agility) as opposed to key technical and tactical actions (e.g., passes, shots, tackles, set pieces etc). However, due to their association with match success [10, 11], coaches routinely prioritise technical and tactical aspects during training sessions [12]. Therefore, for player development and training to be considered truly comprehensive, accurate measures of technical and tactical performance are warranted [13].

Given the rewards associated with players possessing enhanced technical and tactical ability, performance analysis in soccer is regarded as a vital constituent in understanding the demands of academy soccer [14, 15]. To quantify, assess, and improve both individual and team performance, soccer teams routinely use event-based (shots, passes, tackles) key performance indicators [16]. that are commonly retrospectively identified within video footage to inform practice [17–20]. Methods of performance analysis have recently evolved from being reliant on time-consuming notational analysis to the use of advanced technology and automated systems capable of effortlessly collecting and processing large quantities of real-time data [21–23]. Accordingly, utilising technology can help to overcome the constraints faced by manual methods of data collection [24].

Implementing technology into training is thought to increase the precision of the feedback provided to players, whilst also improving performance by assisting to identify optimal techniques and training methods [25–27]. Accordingly, wearable technology is being increasingly employed to assess performance in a variety of team sports including soccer [28] Rugby League [29], Rugby Union [30] and cricket [31]. Furthermore, as more recent developments permit movement performance to be evaluated and monitored, this has enabled the assessment of skill specific performance in soccer [32–34]. Despite the purported promise of wearable

technology, concerns regarding their use and reliability are still prevalent [35–37]. Effective implementation into sporting environments is imperative as poor integration can cause long-lasting ramifications such as practitioners developing negative perceptions of potential utility, in turn creating a reluctance to use wearable technology [35–37].

While there is a plethora of performance analysis research providing guidelines for implementing technology [18, 21, 22, 38, 39], there is a tendency for processes to be oversimplified into flow charts and schemas that neglect to consider the widely acknowledged contextual challenges inherent within soccer [18, 40–43]. The apparent reluctancy from soccer clubs to disseminate their reflections, and experiences of the integration process [44] could be explained by practitioners deeming their experiential knowledge as primarily relevant to their own applied settings Accordingly, it is unsurprising that there is uncertainty regarding the most effective manner to integrate new technology into applied practices [45, 46]. However, this information could provide valuable insight into the ways in which different teams, organisations, and practitioners could modify their approaches to meet both their own, and the external needs that they encounter [44].

Greenhough et al., [47] investigated professional soccer, coaches, practitioners, and players perceptions of the use of virtual reality. Their findings highlighted the need to understand the opinions of key stakeholders, particularly regarding the research questions that they deem to be most important for performance. Furthermore, they discussed the importance of identifying what key stake holders view as potential barriers so that these can be alleviated where possible. Similarly, Wylde et al., [24] sought to uncover coaches' perceptions towards using wearable technology. Emphasising the importance of this in enabling coaches concerns to be addressed, which could in turn enhance the desired synergetic relationship between practitioners and coaches. Although these studies provide insight into how technology can be effectively integrated in an applied sport environment. The methodology is limited due to involving surveys that only provided quantitative results.

Through use of interviews [18] provided a more realistic representation of how performance analysis is delivered within soccer. Their findings highlight the complexities that need to be considered by coaches and practitioners, emphasising the need to uncover the effects that technology has on human interactions, when integrating new technology. Furthermore, Luczak et al., [48] also express the importance of gaining deeper qualitative findings. Demonstrating how a team's organisational structure, and culture, impacts opinions and buy in. Consequently, they acknowledge that the appropriate approach to implementing technology would differ based on the organisational goals. Thus, warranting the need to conduct similar research targeted towards uncovering the perceptions of using technology to assist with the monitoring of technical and tactical metrics. Moreover, as the researchers solely conducted interviews, quantitative results were not provided due to the inherent variance in the results. Adopting a mixed methods design consisting of a quantitative survey and a qualitative interview could overcome the limitations of previous research contributing to a deeper understanding of participants opinions and what motivated their responses [49, 50].

Consequently, it has been suggested that supplementary applied and/or case-based research is necessitated to afford real-life insight into how technology is used and implemented within elite sport [44, 47, 51, 52]. Qualitative research that uncovers existent performance problems that are faced by academy soccer clubs is essential to provide insight into the pedagogical underpinning and ways to effectively implement performance analysis technologies within an applied context [41, 44, 46, 51]. Therefore, to ensure practitioner 'buy-in' when implementing performance analysis technology, it is important to understand which facets of performance analysis practitioners consider to be important, whilst identifying specific barriers they may

face when attempting to implement such analyses within an academy environment. Accordingly, this study sought to identify professional soccer practitioners' perceptions of the application of performance analysis technology (including wearables) within a single academy club. Secondary aims were to understand the perceived importance that professional soccer club practitioners place on monitoring technical and tactical player characteristics, current performance analysis practices, and barriers to implementing wearable technology.

## Methodology

### Study design

A sequential, explanatory mixed-method design (Fig 1) through use of a cross sectional survey (Part A) and interview (Part B) [49] was elected for this study to gain an in-depth and comprehensive understanding of the methods of assessment, reporting strategies, and barriers to the technical and tactical monitoring strategies currently used within an English premier league category one academy. Mixed-method designs hold a number of advantages over single-method designs including their ability to address different aspects of a research topic (e.g., exploratory questions) and to allow stronger inferences to be drawn.

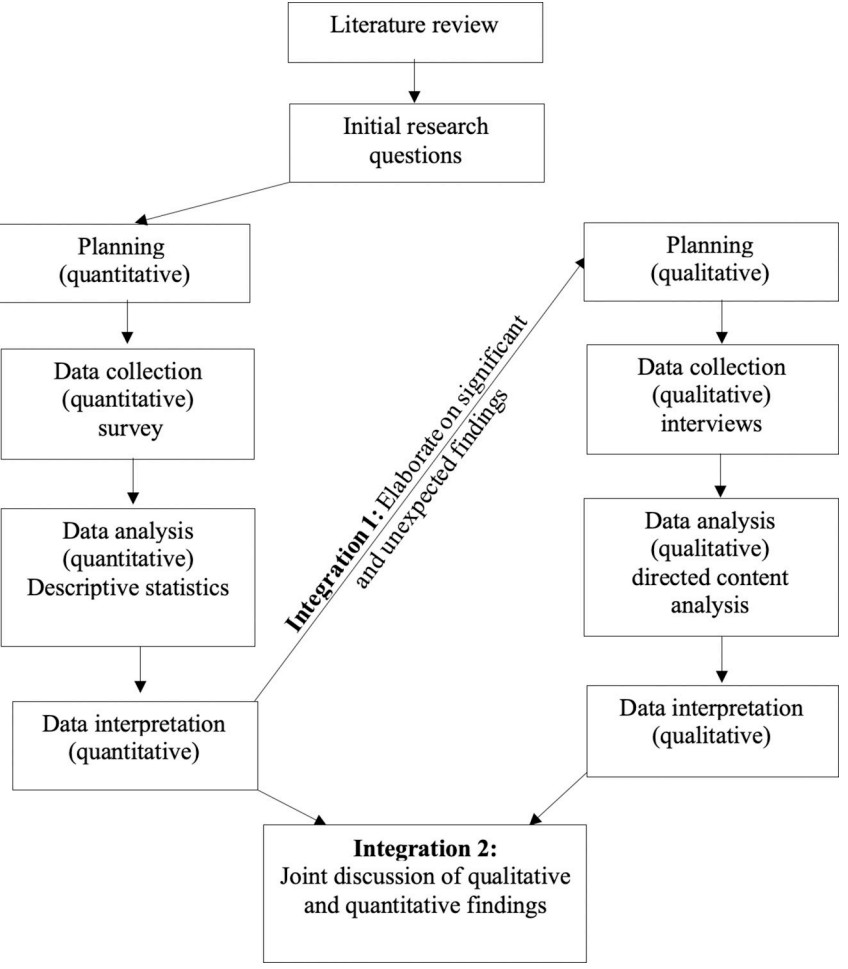

**Fig 1. A schematic representation of the methodology.**

## Part A-Cross sectional survey

**Participants.** Having gained ethical approval by the Faculty of Health Sciences Ethics Committee (University of Hull; FHS423), a purposeful sample of 44 practitioners (male: n = 41; female: n = 3; age = 32 ± 5.8) out of a potential population of 50, who at the time of the study were employed by a specified English Premier League, Category One academy, and directly involved with the monitoring of players, gave written informed consent, and completed the online Microsoft forms closed question (i.e. Yes or No) survey which took approximately 30 minutes to complete. In accordance with previous survey design [13, 53, 54] the opening section of the survey presented respondents with a set of in/exclusion criteria questions including the following: Do you consent to participate?; Have you previously completed (and submitted) this survey?; Are you currently employed to work with players within 'x' Football Club? Failure to meet these criteria prevented practitioners from undertaking the survey and were redirected to the "Thank-you page". Responding practitioners comprised Performance Analysts (n = 12, 27%), Lead Coaches (n = 12, 27%), Assistant Coaches (n = 8, 18%), Sports Scientists (n = 3, 7%), Physiotherapists (n = 4, 9%), Scouts (n = 1, 2%) and practitioners with non-specific managerial roles (n = 4. 9%), who fulfilled positions within the Foundation Phase (Under [U] 9 to U11: n = 8, 18%), Youth Development Phase (U12 to U16: n = 15, 34%), Professional Development Phase (U17 to U21: n = 9, 20%), First Team (n = 7, 16%) and practitioners who worked across all phases (U9 to U21: n = 5, 11%). Responding practitioners possessed 8.5 ± 6.2 years of experience in their roles and 43 (98%) were employed by the club on a full-time basis. Respondent data were collected between June 2022 to August 2022 as this coincided with both off-season and preseason, as this was considered a suitable time period to ensure optimum response rate of practitioners working within the targeted sample. Eligible practitioners were invited to take part via email, with a follow up email one month later, and a third email being sent in the event of no response one month after the second email. Surveys were reviewed for content validity, lucidity, layout, and grammar in consultation with suitable academic staff (n = 2) who all possessed a relevant PhD and with research experience in soccer.

**Survey structure and content.** The survey structure comprised 5 sections (Section 1: General information; Section 2: Technical metrics; Section 3: Tactical metrics; Section 4: Personal perspectives; Section 5: Conclusion of survey). These sections used a total of 40 closed-ended (e.g., Yes/No and Likert scale) questions which were included to assess practitioners' opinions, attitudes, and behaviours, consequently enabling perceptions to be easily operationalised [55]. All information stated in section 1 of the survey corresponded directly to the respondents. To ensure that respondents were not identifiable, the information they provided was anonymised in the analysis process by assigning the response with a unique responded code. Sections 2–3 examined respondents' perceptions of the current assessment of technical and tactical metrics during training and match-play within the sampled professional soccer academy. Section 4 assessed practitioners' personal opinions on the importance and use of technical and tactical metrics. Lastly, Section 5 concluded the survey by requesting respondents to state their over-arching perceptions on the application of automated ways of measuring technical and tactical metrics and if they would consider using foot-mounted Inertial measurement units (IMUs).

Section 1 comprised 8 multiple choice questions selected to collect demographic details. The information required in this section comprised the primary role that they held at the club (i.e., Lead coach. . .etc.), the age group that they work within (i.e., U9, U10, U11 etc.), duration of experience in their discipline, nature of employment and qualifications held (both professional and academic). Sections 2 to 3 focussed on the assessment of technical and tactical

metrics during training and match-play. Respondents were required to answer multiple choice questions to specify whether technical and tactical metrics are currently monitored during both training and match-play, who is responsible for monitoring them, the frequency in which these are currently monitored, if operational definitions are used, and if between practitioner reliability testing is conducted. As this section intended to ascertain practitioners' knowledge the option of "unknown" was also provided throughout. Likert scales were provided to identify the approach taken to feedback technical and tactical metrics in both training and match-play, how this information is fed back and who it is fed back to (e.g., How is the feedback of tactical metrics delivered [. . .]?—Always, very often, often, sometimes, never, unknown). Further-more, to determine any perceived barriers to monitoring technical and tactical metrics in training and match-play an additional 4-point Likert scale was used (e.g., Please indicate which of the following act as barriers within your club when attempting to monitor/ assess tactical metrics- Major barrier, minor barrier, not a barrier, unknown). Section 4 of the survey addressed the respondent personal opinions on the use of performance analysis technology within the academy via the use of multiple-choice design questions to establish which metrics practitioners deemed to be technical or tactical in nature. Information to ascertain the importance practitioners placed on technical and tactical metrics and the level of importance they placed on rationales for monitoring technical and tactical metrics was collected using a five-point Likert scale (e.g., Please rank your perceived importance on being able to quantify each of the below metrics within training -not at all important, low importance, neutral, important, very important). Additional five-point Likert scales (e.g., The value I place on technical metrics depends on [. . .]- strongly disagree, disagree, neutral, agree, strongly agree) were also included to determine what variables practitioners perceived to alter the importance of monitoring. Furthermore, a series of multiple-choice questions were included to discern practitioners' pref-erences for the reporting and feedback of technical and tactical metrics. Lastly, section 5 of the survey served as a concluding section which included three questions which were designed to indicate whether participants would find an automated way of measuring technical and tactical metrics beneficial and if they would consider using foot-mounted IMUs to measure this. Respondents were required to answer these using either a multiple choice (yes, no) format or a five-point Likert scale (e.g., I believe that it would be beneficial to have an automated way of tracking technical metrics- strongly disagree, disagree, neutral, agree, strongly agree). Finally, the choice to opt into participating in a follow up interview was included here.

## Part B: Semi structured interviews

In line with a sequential approach to mixed methods, the quantitative phase of the study was followed by a qualitative phase. A cornerstone of rigour in mixed methods research is that findings from the quantitative phase are linked to or integrated within the qualitative phase [56]. Consequently, in this second phase, findings from the quantitative phase were used to inform the interview guide and qualitative interviews were used to build upon and add further depth to the findings generated by the survey.

**Participants.** Survey respondents that specified their willingness to participate in an interview were approached via email and informed of the interviews purpose and requirements. All partici-pants were required to have had experience with, or knowledge of, monitoring technical and tacti-cal metrics within an elite youth soccer setting. The sample comprised 11 male practitioners (age = 31.18 ± 5.76; years' experience = 7.36 ± 5.76) currently working within a UK professional soccer academy as either a coach (n = 5, 45%), performance analyst (n = 3, 27%), or academy man-agerial role (n = 3, 27%). All practitioners gave their verbal informed consent prior to commence-ment of the interview. Informed consent was recorded using an audio recording device.

**Interview method.** This study is grounded in a postpositivist paradigm which is conceptualised as having an objectivist epistemology (i.e., universal laws exist but discoveries are only approximations of truth) and critical realist ontology (i.e., whilst researchers might pursue an objective truth, knowledge is fallible as it is shaped by contextual factors). This position had several implications for data collection procedures and analysis. For example, interviews were informed by survey data, current literature and consistent for all participants. Single interviews were undertaken, and data was analysed using a combination of inductive and deductive analysis. Furthermore, the trustworthiness of this process was enhanced via peer debriefing (i.e., members of the research team challenged the primary researcher's initial interpretations of the data) and member-checking (i.e., transcripts were returned to participants to ensure they accurately represented their thoughts).

Interviews took a semi-structured approach lasting 56.88 ± 6.03 minutes. An interview guide that was informed by existing research within the field and responses to Part A were utilised during the interviews to offer a certain amount of structure and ensure that the same thematic approach was consistently applied throughout the interviews [57]. However, the order and phraseology of the questions was altered dependent on the flow of the conversations [58]. Throughout the interviews various types of questions were asked. As per Kvale's [59] recommendations, on commencement of the interview participants were provided with interview context, the purpose of the interview and the opportunity to ask any questions. The interview guide addressed a broad variety of topics including practitioners' rationale for monitoring technical and tactical metrics, their perceived advantages and disadvantages of monitoring technical and tactical metrics and their views on implementing new technology. Focused questions were utilised to follow up on survey responses for example "*Please explain the barriers you've faced when monitoring technical and tactical demands*". Open-ended questions such as "*Tell me about your experiences using [. . .]*" were used to elicit rich descriptions of experiences. Follow up probes or curiosity-drive questions for instance "*Please may you elaborate on why you feel [. . .]*" were used to encourage responses that were more in-depth and detailed [58]. Some detail-oriented probes were used whilst others focused on encouraging elaboration for example "*Please can you give me an example of [. . .]*" or for clarification such as "*I'm not sure what you mean by the term [. . .]*? Questions such as "*I am just going to introduce a new topic of [. . .] now*" were used to re-direct interviewees or once a line of questioning had been exhausted [57]. Clarification probes such as "*Would it be correct to say that you believe [. . .]*" were used to increase clarity [59] and to ensure that the participants' views were being accurately represented. To conclude the interviews participants were asked if they would like to add any additional information that the interview had not covered. Interviews were scheduled in advance and took place in person at the club's training ground. Interviews were conducted in a private room, free from distractions to avoid excessive background noise and allow participants to focus [60–62]. Interviews were recorded using an audio recording device and as per literature recommendations a backup recording device was also used [62]. Interviews were ceased when it became clear that no new findings were emerging. Audio data was removed at the earliest opportunity and destroyed after transcription [60]. Audio recordings were anonymised during transcription and transcribed verbatim. As per recommended procedures the principal researcher listened back to the audio recordings while reading the transcriptions to assure the accuracy of participant responses [60].

**Data analysis.** For part A, survey responses were exported from Microsoft Forms to Microsoft Excel (Microsoft Excel for Mac, Version 16.64, Microsoft 2022). Prior to analysis responses were crosschecked for duplications. Responses were to be excluded if replication was found. Subsequently descriptive statistics (frequencies and percentages) of individual item responses were calculated. Following this, to determine a binary (agree or disagree; important

or not important) interpretation, data for composite component data points (i.e., strongly agree-agree, strongly disagree-disagree; very important-important, not at all important-low importance) were aggregated. In accordance with similar previous research [63–65], and due to the limited, non-parametric categorical data collected, statistical assumptions were violated (i.e., the Pearson's chi-squared test necessitates that for expected frequencies, cell values should be >5 in ≥80% of cells, and ≥1 in 100% of cells) [66]) and therefore unviable. Accordingly, data was reported descriptively.

For part B, following transcription, participants were assigned pseudonyms for the purpose of anonymity [67]. Subsequently, directed content analysis was used to identify themes [68]. Familiarisation involved a careful and close reading of transcribed interviews [69]. Subsequently, line-by-line analysis was conducted and sentences from the transcripts, or 'meaning-units', were assigned a label or phrase that captured the participants' perceptions regarding the monitoring of technical and tactical metrics. Here, a deductive approach was employed by using existing theory and survey findings to assign labels to the segmented text. Furthermore, for the inductive process new codes were allocated to any text that were unable to be categorised by the primary coding system [68]. Codes were then aggregated to form emergent categories (lower-order themes) determined by their contrast other categories and resemblance to each other [69]. Lastly to form higher-order themes, the procedure was repeated. For the purpose of enhancing the trustworthiness of data, peer-debriefing encouraged members of the research team to dispute the initial data interpretations of the primary researcher [70] and to reach a consensus that codes and themes were an accurate representation of participants' views and perceptions. Additionally, themes were independently identified by two of the researchers, who then assumed the role of 'critical friends' to challenge each other's perspectives and interpretations [71].

# Results

## Part A-Cross sectional survey

Response data showed that practitioners believed technical metrics are actively monitored in training sessions (80%) and matches (89%), with similar responses reported for tactical metrics (Training: 64%; matches: 91%). Practitioner response data suggests that performance analysts (technical 82%, tactical 77%), Lead coaches (technical 43%, tactical 61%) and assistant coaches (technical 41%, tactical 55%) are primarily responsible for monitoring procedures and responses showed that monitoring is most typically undertaken daily (technical 59%, tactical 55%) and weekly (technical 25%, tactical 30%). Furthermore, the response data indicated that the most common approaches adopted for monitoring are video review (technical 77%, tactical 66%), event coding (technical 61%, 64% tactical) and external providers (e.g., Opta reports) (technical 30%, tactical 27%). Regarding the provision of feedback on technical and tactical metrics, verbal feedback (technical: always (23%), very often (43%) tactical: always (48%), very often (25%)) and video feedback (technical: always (36%), very often (34%) tactical: always (50%), very often (27%)) were most frequently utilised methods of delivery by practitioners. Furthermore, feedback was most delivered 1 to 1(technical: always (14%), very often (45%) tactical: always (23%), very often (46%)), in position specific groups (technical: always (18%), very often (36%) tactical: always (25%), very often (39%)) or to the whole team (technical: always (27%), very often (30%) tactical: always (25%), very often (39%)). Moreover, results showed standardised operational definitions were not consistently used by all practitioners to describe technical (43%) and tactical (52%) metrics. Likewise, the results demonstrated that most practitioners do not knowingly engage in reliability testing when monitoring technical (14%) and tactical (23%) metrics.

**Table 1. Tabulated data demonstrating distributed barriers to monitoring technical metrics based on the experiences of practitioners.**

| Questionnaire Question | Major barrier | Minor barrier | Not a barrier | Unknown | Aggregated barrier [a] | Rank based on aggregated barrier |
|---|---|---|---|---|---|---|
| *"Which of the following act as barriers when attempting to monitor/assess technical metrics…"* | | | | | | |
| *"Time constraints"* | 34% | 43% | 11% | 11% | 77% | 1 |
| *"Staffing numbers"* | 30% | 34% | 18% | 18% | 64% | 2 |
| *"Resource limitations"* | 20% | 30% | 30% | 20% | 50% | 3 |
| *"Location of match (away)"* | 5% | 34% | 39% | 23% | 39% | 4 |
| *"Players buy-in"* | 5% | 34% | 48% | 14% | 39% | 5 |
| *"Financial budget limitations"* | 9% | 27% | 41% | 23% | 36% | 6 |
| *"Staffing competency"* | 0% | 27% | 52% | 20% | 27% | 7 |
| *"Coach buy-in"* | 2% | 25% | 57% | 16% | 27% | 8 |
| *"Management support"* | 2% | 18% | 52% | 27% | 20% | 9 |
| *"Player age (too young)"* | 2% | 18% | 55% | 25% | 20% | 10 |
| *"Sponsorship agreements"* | 0% | 18% | 36% | 45% | 18% | 11 |
| *"Scientific rational/ justification"* | 5% | 11% | 52% | 32% | 16% | 12 |
| *"Parent/ guardian buy-in"* | 0% | 14% | 50% | 36% | 14% | 13 |
| *"Player age (too old)"* | 5% | 9% | 59% | 27% | 14% | 14 |
| *"Practitioners buy-in"* | 0% | 11% | 66% | 23% | 11% | 15 |
| *"Location of match (home)"* | 0% | 11% | 64% | 25% | 11% | 16 |

[a] Aggregated barrier comprises major barrier plus minor barrier.

Practitioner aggregated response data demonstrated that time constraints (technical (77%), tactical (57%)) and staffing numbers (technical (64%), tactical (47%)) were deemed as being the biggest barriers to monitoring technical (Table 1) and tactical metrics (Table 2).

The distribution of participants perceived classification of metrics is presented in Fig 2. Release velocity (m/s) (95%), leg use (%) (91%) and releases by leg (%) (91%) were identified

**Table 2. Tabulated data demonstrating distributed barriers to monitoring tactical metrics based on the experiences of practitioners.**

| Questionnaire Question | Major barrier | Minor barrier | Not a barrier | Unknown | Aggregated barrier [a] | Rank based on aggregated barrier |
|---|---|---|---|---|---|---|
| *"Which of the following act as barriers when attempting to monitor/assess tactical metrics…"* | | | | | | |
| *"Time constraints"* | 14% | 43% | 25% | 18% | 57% | 1 |
| *"Staffing numbers"* | 11% | 36% | 32% | 20% | 47% | 2 |
| *"Match location (away)"* | 5% | 32% | 43% | 20% | 37% | 3 |
| *"Resource limitations"* | 7% | 23% | 45% | 25% | 30% | 4 |
| *"Staffing competency"* | 0% | 23% | 59% | 18% | 23% | 5 |
| *"Player age (too young)"* | 2% | 18% | 52% | 27% | 20% | 6 |
| *"Players buy-in"* | 0% | 20% | 64% | 16% | 20% | 7 |
| *"Financial budget limitations"* | 7% | 11% | 55% | 27% | 18% | 8 |
| *"Coach buy-in"* | 0% | 16% | 68% | 16% | 16% | 9 |
| *"Scientific rational/ justification"* | 0% | 14% | 59% | 27% | 14% | 10 |
| *"Management support"* | 0% | 11% | 66% | 23% | 11% | 11 |
| *"Parent/ guardian buy-in"* | 0% | 11% | 55% | 34% | 11% | 12 |
| *"Practitioners buy-in"* | 0% | 11% | 66% | 23% | 11% | 13 |
| *"Sponsorship agreements"* | 0% | 9% | 52% | 39% | 9% | 14 |
| *"Match location (home)"* | 0% | 9% | 70% | 20% | 9% | 15 |
| *"Player age (too old)"* | 0% | 7% | 66% | 27% | 7% | 16 |

[a] Aggregated barrier comprises major barrier plus minor barrier.

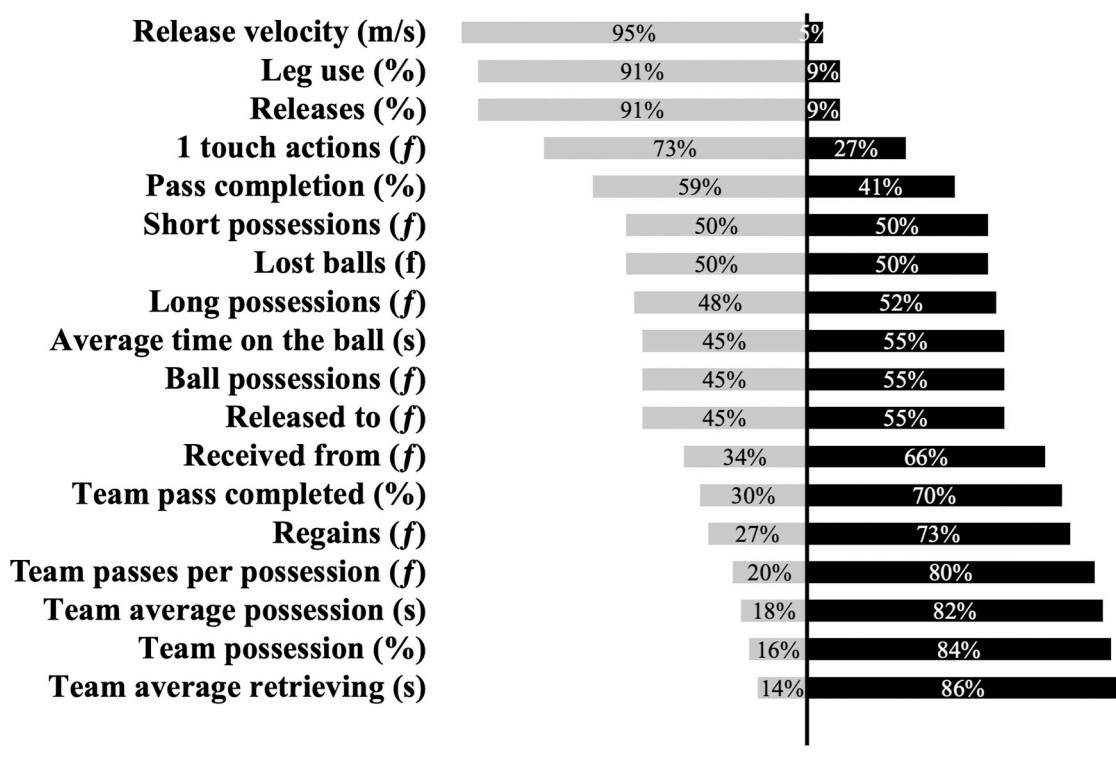

■ Technical  ■ Tactical

**Fig 2. The distribution of practitioner perceived classification of metrics expressed as a percentage.** The grey and black bars correspond to technical and tactical, respectively.

as technical metrics. Whereas, team average retrieving (s) (86%), team possessions (%) (84%) and team average possessions (s) (82%) were identified as tactical metrics. However, the perceived classification of the remaining metrics was unclear. Table 3 demonstrates the distribution of importance practitioners placed on being able to quantify metrics in training and Table 4. In matches. For training sessions, practitioners deemed regains (78%), pass completion (75%) and ball possessions (75%) as the most important metrics to measure. With release velocity (27%), average time on the ball (30%) and releases by leg (39%) being the least important. However, within matches, regains (75%), pass competition (75%) and lost balls (70%) were seen as the most important metrics to measure. For matches, release velocity (25%), releases by leg (34%) and leg use (39%) were shown to be the least important. Furthermore, average time on the ball was considered by practitioners as being 20% more important to monitor in matches than in training, and team average possessions were seen as being 15% more important to monitor in matches opposed to training. Aggregated practitioner results demonstrated that player age (technical (66%), tactical (71%)), player maturation status (technical (66%), tactical (59%)) and playing position/unit (technical (63%), tactical (59%)) were considered by practitioners as the factors that most influenced the perceived importance that they place on the monitoring of technical (Table 5) and tactical metrics (Table 6). Table 7 displays how the purpose for monitoring effects the distribution of importance practitioners place on monitoring technical and Table 8 tactical metrics. For technical metrics practitioners deemed player feedback (91%), player development (82%) and individualisation of training activities (80%) as the most important purposes for monitoring. The EPPP legislation (14%), club legislation (25%) and injury prevention (38%) were the least important. For tactical metrics player

**Table 3. Tabulated data demonstrating the distributed importance practitioners place on being able to quantify metrics in training.**

| Questionnaire question | Not at all important | Low importance | Neutral | Important | Very important | Aggregated not important [a] | Aggregated important [b] | Rank based on aggregated importance |
|---|---|---|---|---|---|---|---|---|
| *"What is your perceived importance on being able to quantify the following metrics within training..."* | | | | | | | | |
| "*Regains* (n)- the number of times a player gained possession straight from the opposition" | 2% | 5% | 16% | 48% | 30% | 7% | 78% | 1 |
| "*Pass completion*- the % of completed passes from player's total passes" | 0% | 11% | 14% | 39% | 36% | 11% | 75% | 2 |
| "*Ball possessions* (n)- the number Possessions a player had control of the ball" | 0% | 9% | 23% | 48% | 20% | 9% | 68% | 3 |
| "*Lost balls* (n)- all possessions that ended with opponent possession or ball out of play" | 2% | 11% | 23% | 41% | 23% | 13% | 64% | 4 |
| "*Team average retrieving* (s)- the average time it took a team to retrieve the ball from the opposition" | 5% | 7% | 23% | 48% | 18% | 12% | 64% | 5 |
| "*Team pass completed*- % of completed passes for the team" | 5% | 5% | 27% | 36% | 27% | 10% | 63% | 6 |
| "*Team passes per possession* (n)- average number of passes per team possession" | 5% | 9% | 25% | 48% | 14% | 14% | 62% | 7 |
| "*Released to* (n)- the number of times a player released the ball and it ended with a teammate receiving" | 5% | 2% | 32% | 43% | 18% | 7% | 61% | 8 |
| "*Short possessions* (n)- number of short possessions a player had (minimum 2 touches possession and duration is lower than 1.5 seconds)" | 2% | 5% | 36% | 43% | 14% | 7% | 57% | 9 |
| "*1 touch* (n)- number of 1 touch actions" | 0% | 7% | 39% | 32% | 23% | 7% | 55% | 10 |
| "*Long possessions* (n)- number of long possessions a player had (minimum 2 touches possession and duration is higher than 1.5 seconds)" | 0% | 7% | 39% | 41% | 14% | 7% | 55% | 11 |
| "*Team possession*- the % of time a team was in possession of the ball" | 5% | 7% | 34% | 30% | 25% | 12% | 55% | 12 |
| "*Received from*- the number of times a player received the ball from another teammate" | 5% | 9% | 36% | 32% | 18% | 14% | 50% | 13 |
| "*Team average possession* (s)- average time of team possession" | 5% | 14% | 32% | 39% | 11% | 19% | 50% | 14 |
| "*Leg use*- the % of touches with each foot" | 7% | 18% | 32% | 25% | 18% | 25% | 43% | 15 |
| "*Releases by leg*- the % of passes/kicks with each foot" | 9% | 16% | 36% | 23% | 16% | 27% | 39% | 16 |
| "*Average time on the ball* (s)- the average time player spent on the ball" | 2% | 14% | 45% | 25% | 14% | 16% | 30% | 17 |
| "*Release velocity* (m/s)- the speed of the foot during the impact with the ball" | 18% | 23% | 32% | 16% | 11% | 41% | 27% | 18 |

[a] Aggregated not important comprises not at all important plus low importance.

[b] Aggregated important comprises important plus very important.

feedback (86%), coach feedback (82%) and individualisation of training activities (82%) were seen as the most important purposes for monitoring. The EPPP legislation (19%), injury prevention (25%) and club legislation (27%) were the least important. Frequency distributions of results showed that visual representations would be practitioners preferred means of reporting

**Table 4. Tabulated data demonstrating the distributed importance practitioners place on being able to quantify metrics in matches.**

| Questionnaire question | Not at all important | Low importance | Neutral | Important | Very important | Aggregated not important [a] | Aggregated important [b] | Rank based on aggregated importance |
|---|---|---|---|---|---|---|---|---|
| *"What is your perceived importance on being able to quantify the following metrics within matches..."* | | | | | | | | |
| *"Regains (n)- the number of times a player gained possession straight from the opposition"* | 2% | 2% | 20% | 48% | 27% | 4% | 75% | 1 |
| *"Pass completion- the % of completed passes from player's total passes"* | 2% | 7% | 16% | 48% | 27% | 9% | 75% | 2 |
| *"Lost balls (n)- all possessions that ended with opponent possession or ball out of play"* | 2% | 5% | 23% | 43% | 27% | 7% | 70% | 3 |
| *"Released to (n)- the number of times a player released the ball and it ended with a teammate receiving"* | 2% | 7% | 20% | 50% | 20% | 9% | 70% | 4 |
| *"Team possession- the % of time a team was in possession of the ball"* | 2% | 9% | 18% | 50% | 20% | 11% | 70% | 5 |
| *"Team average retrieving (s)- the average time it took a team to retrieve the ball from the opposition"* | 2% | 7% | 25% | 45% | 20% | 9% | 65% | 6 |
| *"Ball possessions (n)- the number Possessions a player had control of the ball"* | 5% | 9% | 23% | 50% | 14% | 16% | 64% | 7 |
| *"Team pass completed- % of completed passes for the team"* | 2% | 9% | 27% | 39% | 23% | 11% | 62% | 8 |
| *"Received from (n)- the number of times a player received the ball from another teammate"* | 2% | 7% | 30% | 43% | 18% | 9% | 61% | 9 |
| *"Team passes per possession (n)- average number of passes per team possession"* | 2% | 7% | 32% | 45% | 14% | 9% | 59% | 10 |
| *"1 touch (n)- number of 1 touch actions"* | 0% | 11% | 32% | 39% | 18% | 11% | 57% | 11 |
| *"Short possessions (n)- number of short possessions a player had (minimum 2 touches possession and duration is lower than 1.5 seconds)"* | 2% | 11% | 30% | 43% | 14% | 13% | 57% | 12 |
| *"Team average possession (s)- average time of team possession"* | 2% | 9% | 34% | 39% | 16% | 11% | 54% | 13 |
| *"Average time on the ball (s)- the average time player spent on the ball"* | 2% | 9% | 39% | 36% | 14% | 11% | 50% | 14 |
| *"Long possessions (n)- number of long possessions a player had (minimum 2 touches possession and duration is higher than 1.5 seconds)"* | 2% | 11% | 36% | 34% | 16% | 13% | 50% | 15 |
| *"Leg use- the % of touches with each foot"* | 7% | 23% | 32% | 25% | 14% | 30% | 39% | 17 |
| *"Releases by leg- the % of passes/kicks with each foot"* | 9% | 16% | 36% | 23% | 11% | 27% | 34% | 16 |
| *"Release velocity (m/s)- the speed of the foot during the impact with the ball"* | 23% | 20% | 32% | 16% | 9% | 43% | 25% | 18 |

[a] Aggregated not important comprises not at all important plus low importance.

[b] Aggregated important comprises important plus very important.

for technical (39%) and tactical (39%) metrics followed by video feedback (Technical (25%), tactical (34%)). Results also demonstrated that practitioners would prefer to deliver feedback on technical metrics to players on a 1 to 1 basis (75%), and as a whole team (48%) or on a unit basis (41%) for tactical metrics Practitioner aggregated results indicated that the majority of

**Table 5. Tabulated data demonstrating how contexts effect the distributed value practitioners place on technical.**

| Questionnaire question | Strongly disagree | Disagree | Neutral | Agree | Strongly agree | Aggregated disagree [a] | Aggregated agree [b] | Rank based on aggregated agree |
|---|---|---|---|---|---|---|---|---|
| *"The value that I place on technical metrics depends on. . ."* | | | | | | | | |
| *"Player age"* | 7% | 9% | 18% | 36% | 30% | 16% | 66% | 1 |
| *"Player maturation status"* | 9% | 9% | 16% | 43% | 23% | 18% | 66% | 2 |
| *"Playing position/unit"* | 9% | 14% | 14% | 43% | 20% | 25% | 63% | 3 |
| *"Match context (winning)"* | 16% | 20% | 32% | 20% | 11% | 36% | 31% | 4 |
| *"Match context (losing)"* | 25% | 23% | 20% | 23% | 9% | 48% | 32% | 5 |
| *"Weather conditions"* | 27% | 27% | 16% | 20% | 9% | 54% | 29% | 6 |
| *"Match context (drawing)"* | 25% | 23% | 30% | 14% | 9% | 48% | 23% | 7 |
| *"Match location (home)"* | 25% | 30% | 27% | 14% | 5% | 55% | 19% | 8 |
| *"Match location (away)"* | 25% | 30% | 27% | 14% | 5% | 55% | 19% | 9 |

[a] Aggregated disagree comprises of disagree plus strongly disagree

[b] Aggregated agree comprises agree plus strongly agree

practitioners believe that it would be beneficial to have an automated way of tracking technical (79%) and tactical (71%) metrics. Similarly, results showed that most practitioners would consider using a foot-mounted IMU to monitor technical (68%) and tactical (57%) metrics.

## Part B-Interviews

Interviews highlighted that the monitoring of technical and tactical metrics was viewed favourably by most practitioners. However, time constraints and staffing numbers impact the opportunity to frequently and objectively monitor players technical and tactical performance. Subjective information is predominantly used to assess players technical and tactical performances. However, practitioners see the benefits of using wearable technology to aid with the

**Table 6. Tabulated data demonstrating how contexts effect the distributed value practitioners place on tactical metrics.**

| Questionnaire question | Strongly disagree | Disagree | Neutral | Agree | Strongly agree | Aggregated disagree [a] | Aggregated agree [b] | Rank based on aggregated agree |
|---|---|---|---|---|---|---|---|---|
| *"The value that I place on tactical metrics depends on. . ."* | | | | | | | | |
| *"Player age"* | 5% | 11% | 14% | 39% | 32% | 16% | 71% | 1 |
| *"Playing position/ unit"* | 5% | 18% | 18% | 43% | 16% | 23% | 59% | 2 |
| *"Player maturation status"* | 9% | 16% | 16% | 34% | 25% | 25% | 59% | 3 |
| *"Match context (drawing)"* | 16% | 20% | 34% | 23% | 7% | 36% | 30% | 4 |
| *"Match context (losing)"* | 18% | 20% | 32% | 25% | 5% | 38% | 28% | 5 |
| *"Match context (winning)"* | 25% | 23% | 25% | 18% | 9% | 48% | 27% | 6 |
| *"Match location (away)"* | 18% | 30% | 27% | 16% | 9% | 48% | 25% | 7 |
| *"Weather conditions"* | 23% | 27% | 25% | 20% | 5% | 50% | 25% | 8 |
| *"Match location (home)"* | 23% | 30% | 27% | 14% | 7% | 53% | 21% | 9 |

[a] Aggregated disagree comprises of disagree plus strongly disagree

[b] Aggregated agree comprises agree plus strongly agree

**Table 7. Tabulated data demonstrating how the purpose for monitoring effects the distribution of importance practitioners place on technical metrics.**

| Questionnaire question | Not at all important | Low importance | Neutral | Important | Very important | Aggregated not important [a] | Aggregated important [b] | Rank based on aggregated important |
|---|---|---|---|---|---|---|---|---|
| **"What is you perceived level of importance on monitoring technical metrics for the purpose of…"** | | | | | | | | |
| "Player feedback" | 0% | 0% | 9% | 57% | 34% | 0% | 91% | 1 |
| "Overall player development" | 0% | 5% | 14% | 32% | 50% | 5% | 82% | 2 |
| "Individualisation of training activities" | 0% | 0% | 20% | 43% | 36% | 0% | 80% | 3 |
| "Evaluating match play" | 0% | 7% | 16% | 48% | 30% | 7% | 78% | 4 |
| "Coach feedback" | 0% | 7% | 18% | 50% | 25% | 7% | 75% | 5 |
| "Evaluating training performance" | 0% | 9% | 18% | 45% | 27% | 9% | 72% | 6 |
| "Prescription of future training activities" | 0% | 5% | 25% | 43% | 27% | 5% | 70% | 7 |
| "Talent identification/ player recruitment" | 2% | 9% | 25% | 23% | 41% | 11% | 64% | 8 |
| "Return to play monitoring" | 5% | 7% | 25% | 43% | 20% | 12% | 63% | 9 |
| "Systematic progression of training through age groups" | 2% | 11% | 27% | 27% | 32% | 13% | 59% | 10 |
| "Training load monitoring" | 7% | 16% | 18% | 41% | 18% | 23% | 59% | 11 |
| "Player retention decisions" | 9% | 9% | 25% | 36% | 20% | 18% | 56% | 12 |
| "Injury prevention" | 7% | 16% | 39% | 18% | 20% | 23% | 38% | 13 |
| "Club legislation" | 9% | 20% | 45% | 18% | 7% | 29% | 25% | 14 |
| "Elite Player Performance Plan legislation" | 16% | 18% | 52% | 9% | 5% | 34% | 14% | 15 |

[a] Aggregated not important comprises not at all important plus low importance

[b] Aggregated important comprises important plus very important

collection of objective data. Three main themes were identified which portray the practitioners' views on the monitoring of technical and tactical performance and the implementation of wearable technologies to aid with this. These were: 1) Facilitating player development; 2) The need for objective data to be supplemented with contextual information and 3) barriers to the successful implementation of monitoring technology and recommendations for increased uptake.

**Facilitating player development.** Although results showed that subjective opinion often informs decision-making practitioners acknowledged the benefit of supplementing this with objective measurements to assess technical and tactical performance. In accordance with part A (i.e., survey results), they considered that having objective measurements would have considerable utility to support their subjective opinions, assess the effectiveness of training sessions and enable the individualisation of training activities.

**Assess the effectiveness of training sessions.** The interviews indicated that practitioners considered that having objective data on technical and tactical performance would be advantageous in assessing the effectiveness of their training sessions. When discussing how they would use objective measures of technical and tactical performance William stated that:

> I think you can maybe measure the [technical and tactical] concepts and principles in terms of how you want to play and whether you're implementing those in the session.

**Table 8. Tabulated data demonstrating how the purpose for monitoring effects the distribution of importance practitioners place on tactical metrics.**

| Questionnaire question | Not at all important | Low importance | Neutral | Important | Very important | Aggregated not important [a] | Aggregated important [b] | Rank based on aggregated important |
|---|---|---|---|---|---|---|---|---|
| *"What is you perceived level of importance on monitoring tactical metrics for the purpose of..."* | | | | | | | | |
| *"Player feedback"* | 0% | 2% | 11% | 50% | 36% | 2% | 86% | 1 |
| *"Coach feedback"* | 2% | 5% | 11% | 48% | 34% | 7% | 82% | 2 |
| *"Individualisation of training activities"* | 0% | 7% | 11% | 55% | 27% | 7% | 82% | 3 |
| *"Evaluating training performance"* | 0% | 11% | 9% | 48% | 32% | 11% | 80% | 4 |
| *"Evaluating match play"* | 0% | 5% | 16% | 39% | 41% | 5% | 80% | 5 |
| *"Overall player development"* | 2% | 2% | 16% | 52% | 27% | 4% | 79% | 6 |
| *"Prescription of future training activities"* | 0% | 2% | 27% | 45% | 25% | 2% | 70% | 7 |
| *"Talent identification/ player recruitment"* | 5% | 18% | 18% | 30% | 30% | 23% | 60% | 8 |
| *"Systematic progression of training through age groups"* | 0% | 7% | 36% | 39% | 18% | 7% | 57% | 9 |
| *"Player retention decisions"* | 14% | 7% | 25% | 36% | 18% | 21% | 54% | 10 |
| *"Training load monitoring"* | 5% | 18% | 25% | 39% | 14% | 23% | 53% | 11 |
| *"Return to play monitoring"* | 9% | 14% | 27% | 34% | 16% | 23% | 50% | 12 |
| *"Club legislation"* | 11% | 23% | 39% | 16% | 11% | 34% | 27% | 13 |
| *"Injury prevention"* | 18% | 25% | 32% | 18% | 7% | 43% | 25% | 14 |
| *"Elite Player Performance Plan legislation"* | 18% | 18% | 45% | 14% | 5% | 36% | 19% | 15 |

[a] Aggregated not important comprises not at all important plus low importance

[b] Aggregated important comprises important plus very important

Practitioners want to use the objective technical and tactical data to enable them to understand how players are performing, the reasons why, and to consequently develop players based on this. Equally, Alfie discussed how he would use objective measures of technical and tactical performance to develop players:

> It would just be used to guide decision making or to support like findings or even like to contradict things if we thought things were going to certain way with the player. So, for example if we're assessing ball striking the player might be really good at right foot passing. But [when looking at the technical and tactical measures] we are finding it is not necessarily as we thought...

For Alfie and many of the practitioners, objective technical and tactical metrics allow them to identify problems that require intervention.

Furthermore, practitioners would like to assess whether training sessions are providing their intended technical and tactical loading and would then use that information to aid with designing future training sessions. To illustrate, Keith said:

> I think training sessions should be based upon the principles that you want the players to deliver at the weekend, whether that is technical or tactical and therefore training should

reflect that. And the only way to know whether the training is actually reflecting what's happening in a game is to measure a game technically, tactically to see if then there is a difference.

Practitioners value monitoring systems that allows them to assess the effectiveness of their training sessions by comparing the players technical and tactical outcomes from training to matches.

**Understanding role and player specific needs.**   It was apparent from the interviews that practitioners deemed objectively monitoring technical and tactical metrics as essential for understanding playing position specific demands and to enable training sessions to be tailored to the individual needs of players and aid with their progress and development. For example, William detailed:

. . .you can look at [technical and tactical metrics] over a longitudinal period to track progression. Then we would link in with how they've been training and whether or not we do enough individual work with this player. So, if he's one of the lowest compared to other players of his position, we would look to see if we have done many training sessions around this area with this individual or any sessions or analysis.

This demonstrates how practitioners view having objective measures of technical and tactical performance as beneficial in aiding with understanding the role specific requirements for players, monitoring players individual progress and understanding their development and training exposure needs. Similarly, Colin saw the benefits to objectively understanding players technical and tactical performance:

I think it would help from a match point of view. . .I think it's also a marker that can identify areas where we need to expose younger players to more. . . use the data that we've got to say well, players that play in this position get it a lot more there. So, let's move him there to practice that.

This is evidence of practitioners wishing to use technical and tactical metrics to check and challenge their initial views of players match performances. Furthermore, demonstrating how practitioners would utilise the objective measures of performance to individualise training and meet players development needs.

**Context.**   Findings revealed that practitioners viewed the monitoring of technical and tactical metrics positively. However, their current practice is heavily reliant on subjective opinion when seeking to understand and measure these metrics. These subjective opinions are predominantly based on several contextual factors which include, the individual, session design, and the level of opposition faced.

**Understanding the individual.**   With regards to utilising objective measures on technical and tactical performance, practitioners discussed the importance of including this context when interpreting and seeking to understand what the data might mean, and how this impacts the value they place on those results. It was apparent that practitioners see the benefit of combining objective measures with subjective opinions. However, they strongly felt that they should not be used in isolation or replace their understanding of the player. There were multiple reasons evidenced including Williams's argument that:

I think we should never take away the eye of the coach. As well and being careful because the interactions of a coach and their personality, and the way they are around the people is

miles more important than all the technical and tactical stuff. . . There's no one right approach it's based on that individual player and getting the best out of them.

There were multiple examples of practitioners combining objective and subjective information to tailor their approaches based on their knowledge of an individual and what works best for them. Equally, Colin discussed how:

You are dealing with people. . . Like they are not AI. They are not programmed; they are the not all the same either. . . So, I've got players in my group, who once they put the foot-mounted IMUs on will become obsessed with their numbers. I've got players, couldn't care less. It still doesn't mean I don't use it; it's just how I use it to get the best out of player A or player B. and that's the thing with any kind of technology is that it can't replace this human element of discussion.

**Statistics do not capture everything.**   Practitioners also believed that objective data should be combined with subjective information as statistics alone cannot measure every aspect of soccer performance. For example, Dave explained how:

If you're looking at stats and if people see that as the only thing, and work solely based off stats. Then they might be doing the player a disservice. Because some things ultimately won't get picked up from stats.

Furthermore, looking at numerical data and statistics alone can be misleading, causing incorrect decisions to be made. According to Alistair:

I just think it's hard to measure some parts of our game, yeah some bits like especially the tactical stuff so hard to quantify so for example if you were like (video) coding and the player gives the ball away and clears it for example on build up that would be noted down as him giving the ball away, but that could have actually have been a really high pressure moment where we were about to concede so him clearing the ball, that's actually a really good decision even though he's given the ball away which would stats wise go down as a bad decision.

These findings demonstrate that in order to get a comprehensive picture of performance practitioners believe objective data should be coupled with subjective information.

**Barriers to the successful implementation of monitoring technology and recommendations for increased uptake.**   Whilst practitioners were, overall, favourably disposed towards using technology systems to monitor technical and tactical performance, they noted a number of factors that either limit the efficacy of these systems or which make it difficult to embed them within training programs. These barriers were principally related to time constraints and staffing numbers, a desire not to over monitor, levels of understanding of the data, the need for operational definitions, and practitioner's understanding of the rationale for monitoring metrics.

**Time constraints and staffing numbers.**   In line with survey results, time constraints and staffing numbers were seen as the main barriers to monitoring technical and tactical performance metrics within an academy.

As demonstrated by Keith's views:

The biggest barrier to monitoring. . . It's usually resource, whether that's financial, to get all the equipment that you need to monitor or human resources in terms of the amount of people that you need to capture it and then process it.

Constraints include not only the time needed to collect the data, but also the time required afterwards to analyse and disseminate it appropriately. When discussing this challenge Derek noted:

> It usually requires some level of manual either input, entry, evaluation or just discussion. I think when someone has to do something extra in an already busy world. Then that can be perceived as negative, albeit for the positives it might bring . . . so that's why I think you have to carefully position it with what it could help reduce.

This reveals some of the challenges that can be faced when attempting to introduce new metrics or trying to integrate a new technology into practice. Even when the metrics provided may be of benefit, those introducing them must be mindful of the time pressure this puts on practitioners and or players. Therefore, any new methods of data collection and technologies need to both add important information and streamline processes for them to be successful. However, although practitioners appreciate how technology can streamline processes and reduce workload, they also acknowledge the time required to become accustomed to using it. This is perhaps best summarised by Alistair who noted that:

> Being up-skilled with like software and stuff like that it takes a while and when we are so busy it's hard to fully get up-skilled so like sometimes just knowing how to use the software [is a challenge].

These findings demonstrate how time constraints can affect practitioners' ability to confidently use new technologies and understand and apply the data being provided by them.

**Over monitoring.**  When discussing barriers faced with monitoring players technical and tactical metrics it was clear from practitioners that there was a concern regarding over monitoring what the players are doing. For example, Tommy stated:

> If everything we did every day was measured. . .I think it might affect how we would work naturally. So, if we measure too many things or the individuals, I think it could potentially stop certain individuals from being who they really want to be.

This demonstrates how practitioners are conscientious about of the ethical implications of excessive monitoring conducting good practice and taking the individuals into account throughout monitoring processes. Colin echoes the importance of informing and involving players within the monitoring processes:

> [Players] can try and impress the coach and do things that they wouldn't naturally do to for example, show that they can play left and right, so they might have a 50:50 split of left and right technical actions, but their performance has fell through the floor because they are focusing on the wrong things. So that's the only barrier I can think of.

Practitioners felt that over monitoring performance metrics without explanation to players can cause them to alter their behaviours negatively and focus on improving their performance on specific metrics rather than performing context-appropriate actions. Additionally, practitioners reflected about the quantity of performance data currently collected on players. Alistair stated that:

> You can run the risk of information overload and it not actually being worth the money invested in getting the data. I guess it's just additional information that we can share with players to help them. However, there's a fine balance, because we already have a lot of information.

Here we can see that practitioners are mindful about prioritising what metrics are monitored by only collecting data if it is conducive to improving players performance.

**Understanding the data.**   It was apparent from the interviews that for successful implementation and adherence to collecting technical and tactical metrics coaches require a clear understanding of what the provided data shows and means for performance. For example, Keith explains some of the potential challenges to integrating objective metrics of technical and tactical performance:

> It could be, you know, the [coaches] comfort levels with data. They might not have been around data much during their coaching careers or if they were players previously during their playing careers. It might be, you know, fear factor of being exposed cause maybe they don't understand it.

This is an example of how confidence levels around using data can have an impact on successful implementation. As a result, insights need to be clear and accurate and delivered in a meaningful way to coaches and players. Furthermore, a rationale and justification for the use of metrics needs to be provided. Moreover, Edward discusses the need to recognise what the technical and tactical metrics mean, which are important and the appropriateness of who and when to feed them back:

> So sometimes you can kind of get a little bit lost in the stats, whereas I think. It's how much stats you want for yourself, for one, but then also how much you wanna relay back to your players, if at all. So. . .the easiest thing would probably be just to give every player every access to all the stats, but actually it's not. . .it's not relevant and it's not appropriate for them, so it's just picking out what you need at which moments.

**Operational definitions.**   Interviews disclosed the need to have clear operational definitions of what performance metrics are being measured. When considering monitoring technical and tactical performance William deliberated:

> I think it's hard cause it's quite it can be quite subjective as well opinion based in terms of what you believe in the moment if it is right or wrong or successful or unsuccessful. . .and everyone sees the game differently which is part of parcel football, so I think yeah if you're looking to measure it, I think you have clear and concise like descriptions of what the metrics are and if not it becomes quite cloudy.

This emphasises how practitioners see the need to for clear operational definitions when assessing players technical and tactical performance to increase the reliability and accuracy of data collection across practitioners and contexts. Keith further reiterated the need for operational definitions, maintaining that they were essential:

> . . .you need clarity around. For example, what is a tactical metric? What is a shot on target? What's a shot off target? Does hitting the post count? Etc. you know, really obvious things like that, but making sure that everybody absolutely understands what the data is saying, what it's reflecting. What it means? And so, people then can't misinterpret what the data is telling us.

He went on to explain the problems that not having operational definitions can cause:

Ohh huge huge amounts of confusion, huge amounts of mistrust. People don't trust the data; they don't understand the data and then people lose confidence in their ability to use it because they don't understand it.

These findings show how operational definitions can minimise confusion, increase practitioners trust in the data and aid with driving meaningful insights.

Furthermore, Derek spoke around the importance of distilling the key requirements of performance from coaches stating:

In order to make the best use of all the different expertise you might have in the support team. Knowing what they know or knowing what they're trying to develop is really important to yeah make the best of what you have got in your team.

Regarding recommendations for uncovering coaches' knowledge of key performance requirements he suggested:

In the ideal world, you'd have all the coaches in a room for two days and you would just map out and just distil, in a hopefully a reasonable way and try to pull out all that information, galvanizing it into a model or a framework or like philosophy of how we do things. That would be, I think, a big step.

This further demonstrates the requirements for having clear operational definitions, and how this could ensure that everyone is working efficiently and effectively towards the same goals.

**Informed of relevance and rationale.** Interviews revealed that in order to gain buy-in when introducing a new technology, it is necessary to inform both practitioners and players of the rationale for its use and then subsequently provide them with feedback afterwards. When considering what would create buy in for implementing new technology Keith explained:

I would feel receptive to using technology if I knew what question, what performance question was being answered by the technology, or what we think the technology will help us answer, why we think it's going to make us better, the team better, the staff better.

Illustrating that practitioner's main focus is on improving players performances. Consequently, having a clear rationale for use, tangible outcomes and continued feedback from new technology will aid with implementation.

Similarly, it was identified that for successful implementation of technology players also need to be informed of the rationale for its use. For instance, Bill said:

For the players again, relevance. Do they think it's going to be relevant to their performance? is it important for how they're going to develop? And education for them to actually sit down and say this is why we're doing it. Because I think sometimes that's missed and the players don't understand why they're actually doing something a certain way.

## Discussion

The purpose of this study was to identify the perceptions of professional soccer practitioners' regarding the application of performance analysis technologies (including wearables) within a single academy environment. This included understanding the perceived importance that

professional soccer club practitioners place on monitoring technical and tactical player characteristics, current performance analysis practices, and barriers to implementing wearable technology. The main findings of this mixed-method study were four-fold: (1) it is evident that within current academy practices technical and tactical metrics are monitored more frequently in matches than training and that time constraints and staffing numbers were viewed as being the predominant reason for this with the majority of practitioners believing that an automated way (e.g. foot-mounted IMU's) of tracking technical and tactical metrics would be beneficial; (2) there is a consensus amongst practitioners that monitoring technical and tactical metrics in an academy setting is beneficial to assist with player development and enrich feedback provision, however, (3) practitioners are careful not to over-monitor players performance. Lastly, (4) for successful implementation and continued uptake the information delivered from monitoring technical and tactical performance needs to be meaningful to players and practitioners and be able to be easily understood and interpreted.

The finding that technical and tactical metrics are monitored less in training than in matches mirrors previous literature. For example, Marris et al., [72] discussed that despite their importance for performance, technical metrics are neglected repeatedly by practitioners owing to time constraints, staffing numbers and finances. This is in keeping with the most prevalent barriers to monitoring reported in the present study (Table 1). Accordingly, Roell et al. [73] have called for more appropriate methods of data collection that can be easily applied within training sessions. One potential solution involves the use of wearable technology, which has been commonly employed to assess players performance and mitigate injuries in team sports [28, 48]. Although the majority of commercial IMUs are not designed to specifically quantify team sport related technical/tactical metrics, foot-mounted IMUs may now be used to assess skill specific performance in soccer [32, 33]. Accordingly, the use of foot-mounted IMUs to evaluate and monitor movement performance could be extended as a skill specific coaching aid [34]. Wearable IMUs might serve as a potential solution to the time and labour-intensive monitoring processes [24] faced by practitioners in the current study.

Practitioners viewed using technology to aid with the monitoring of technical and tactical metrics in a positive light, particularly regarding its benefit in assisting with player development and enriching feedback provision. This could be explained due to the integration of technology within training sessions easing and accelerating feedback delivery, enabling its provision during real-time performance [74, 75]. Furthermore, the use of technology in sport helps to improve performance by helping to identify optimal techniques and methods, whilst also increasing the precision of the results and feedback offered [25–27]. Likewise, electronic tracking devices have been used to evaluate and establish performance norms [28, 76]. Such information can support talent development processes by highlighting performance characteristics and expressing them relative to development norms according to the athlete's age and biological maturation [77, 78]. For instance, Towlson et al., [78] have used foot-mounted IMU's to establish that pitch size has a large effect on technical and tactical metrics, with smaller pitch areas leading to increased technical actions. In addition, the use of foot-mounted IMU's derived player passing network metrics has shown that early maturing players (typically taller, heaver and faster [79, 80] became more integral to passing and team dynamics when playing in a mixed maturation team. Findings like these provide evidence for the application of such technologies in talent development programmes.

The ability to monitor progress not only aids with identifying improvements, and enabling causality to be established, but it can also be used for motivational purposes [81, 82]. Accordingly, it is unsurprising that practitioners in this study demonstrated an openness to adopting the use of wearable technology as they disclosed a strong desire to facilitate player development throughout. Moreover, evidence of practitioners exhibiting an openness to using technical and

tactical metrics to aid with their session design and review is encouraging as this indicates that those responsible for player development are focussed on improving their practice. For example, being able to objectively monitor both physical and skill related behaviour in training and matches has been shown as imperative for quantifying and understanding match demands, developing training programmes, and reducing the probability of non-contact injuries [48, 83].

Regardless of wearable technologies promise or potential, concerns regarding their use are still prevalent [35]. Effective implementation into sporting environments is imperative as poor integration can have long-lasting ramifications. Technology needs to be implemented in a manner that influences and informs practices [51]. This view was reflected by practitioners in the present study explaining their desire to be informed of the relevance and rationale for any monitoring. Furthermore, wearables provide the potential to measure almost every conceivable parameter. However, doing so is not always practical or suitable for athletes [35]. Practitioners were mindful of this and accordingly wanted to be able to tailor information to individual's needs and learning styles. Likewise, they were wary that technology might lead to over-monitoring of players. This may be due to high frequencies of monitoring being seen to burden athletes causing them to focus on performing well on specific metrics in turn initiating behavioural changes that can be of detriment to their performance [84]. Moreover, it has been shown that high-tech solutions and precise collection of data does not inherently equate to collection of meaningful data [51]. Hence practitioners desired objective data to be coupled with context to ensure that the data was meaningful. Furthermore, the appropriateness of adopting technology is dependent on multiple factors such as budget, team strategy and the organisational structure [28]. What constitutes 'correct' data is context and organisation specific. This can depend on the performance questions that are asked by staff members and the way in which an organisation's structure enables decisions to be informed by data [51]. Accordingly, comprehending how organisations perceive athlete data needs to be deliberated and prioritised; should coaches and key stakeholders not engage with the need for data collection challenges may be faced [28]. Key stakeholders need to understand the purpose of data collection, and what data is presently accessible against what data is still required [85, 86]. Thus, substantiating practitioners desire to be informed and involved in the selection and justification of adopting new technology into practices.

Similarly, a failure to understand why data is being collected and how it can be most effectively applied can prove detrimental to athlete-practitioner relationships [87]. To combat this all parties associated with athletes' performance decisions should be acknowledged and consistently engaged with particularly regarding decisions about purchasing new technologies [28]. Likewise, for successful use and implementation coaches need to be provided with feedback, understand the information they are receiving and be able to act on this data [88] to enable the feedback loop to be closed. Accordingly, as evidenced in the results of the present study, without understanding the information provided practitioners will not be able to confidently modify their practices and players performance based on wearable-derived data. Furthermore, barriers to integration have been identified regarding how data is communicated effectively, thus comprising of the timing, frequency and length of feedback, and the approach taken to disseminate insights [87]. A prominent challenge faced is the appropriate interpretation of data and how decision making is consequently affected [89]. This is dependent on the desired use of wearable technology and the environment that it is implemented within. Understanding end-users' needs, interest and intended decision making processes is vital to enable bespoke information to be provided in a clear and concise manner that is easily interpreted [87].

Although this study identifies professional soccer practitioners' perceptions regarding the application of performance analysis technologies within an academy environment, there are

some limitations. For example, it should be acknowledged that the sample sizes included within each part of the study were predominantly small in size. However, this is to be expected as the sample consisted of staff members from a single category one soccer academy, so it was inherently small population. Nonetheless, Hennink & Kaiser [90] discussed how data saturation can be reached in as little as 9 interviews, particularly in a study like the present where the population is relatively homogenous.

Additionally, whilst the intention was always to understand individual perceptions, it is possible that due to respondents being employed by the same club, they are likely to have a shared a club-based philosophy on the use of technology and the importance of monitoring technical and tactical metrics. Consequently, this may have magnified the findings in a particular direction. Accordingly, results from the present study are only indicative of that specific academy, and may not represent the perceptions of all category one academy practitioners within the English premier league. Therefore, further studies with a greater number of high-performance practitioners are warranted; this will likely require collaboration between premiere league category one academies. Likewise, it is recommended that future research could also extend to include academy soccer players perceptions too.

## Conclusion

Through combination of a detailed survey and in-depth interviews, this study has delivered a unique and comprehensive insight into the importance that professional soccer club practitioners place on monitoring technical and tactical player characteristics and the barriers to implementing wearable technology to aid with monitoring within a single academy setting. Accordingly, we believe that this study provides a novel insight and first-hand account of how technical and tactical metrics are being used within a category one premier league academy in both training and match-play, and the ways in which technology can be implemented to aid with this. Crucially for the design of future research and for prospective users of wearable technology who may wish to monitor technical and tactical metrics, this study shares the perceived uses, barriers, and recommendations for successful implementation. Significantly, consensus was displayed amongst practitioners regarding the benefits of monitoring technical and tactical metrics for player development purposes. However, findings revealed that time constraints and staffing numbers hinder their ability to capture these metrics but that using foot-mounted IMUs would aid with automating this information. That said, practitioners are aware of the limitations of monitoring technologies and are cognisant of the dangers associated with the over-monitoring of players. Conclusively, it is suggested that before adopting the use of new technology a needs analysis should first be undertaken. Succeeding this, the implementation of new technology into an academy environment should be a collaborative approach. It is essential that care is given to ensure that the relevance and rationale for monitoring is clearly outlined to all involved. Likewise, when feeding back information, it is imperative that it is both meaningful and easy for coaches to understand and implement. Following these steps would aid with the effective implementation and sustained use of technology to assess technical and tactical performance. Further, enabling the data collected to be applied by coaches to both monitor and positively impact players performances.

## Acknowledgments

The authors would like to thank and acknowledge the survey respondents and interviewees for their contributions. This study would not have been possible without these.

## Author Contributions

**Conceptualization:** John Toner, Chris Towlson.

**Formal analysis:** Tia-Kate Davidson, Chris Towlson.

**Funding acquisition:** John Toner, Chris Towlson.

**Investigation:** Tia-Kate Davidson, John Toner.

**Methodology:** John Toner, Chris Towlson.

**Project administration:** Steve Barrett.

**Software:** Steve Barrett.

**Supervision:** Steve Barrett, John Toner, Chris Towlson.

**Visualization:** Tia-Kate Davidson.

**Writing – original draft:** Tia-Kate Davidson, John Toner.

**Writing – review & editing:** John Toner, Chris Towlson.

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
