## [Decision Letter · Decision Letter 0]

20 Nov 2023

PONE-D-23-34955Professional soccer practitioners’ perceptions of using performance analysis technology to monitor technical and tactical player characteristics within an academy environment.PLOS ONE

Dear Dr. Davidson,

Thank you for submitting your manuscript to PLOS ONE. After careful consideration, we feel that it has merit but does not fully meet PLOS ONE’s publication criteria as it currently stands. Therefore, we invite you to submit a revised version of the manuscript that addresses the points raised during the review process.

We look forward to receiving your revised manuscript.

Kind regards,

Filipe Manuel Clemente, PhD

Academic Editor

PLOS ONE

Journal Requirements:

Additional Editor Comments:

Thank you for your submission, Reviewer #1 appreciates the relevance of your study, commending the clear methodology and results. However, they suggest expanding the introduction, clarifying basic concepts, and refining certain sections. Reviewer #2 acknowledges the strong introduction but recommends a deeper exploration of previous research and clarifying the study's focus. Both reviewers advise on improving data visualization and addressing potential bias in selecting a single academy. Please incorporate these suggestions to enhance the manuscript's clarity and comprehensiveness. We look forward to receiving your revised submission.

Reviewers' comments:

Reviewer's Responses to Questions

**Comments to the Author**

1. Is the manuscript technically sound, and do the data support the conclusions?

Reviewer #1: Yes

Reviewer #2: Yes

2. Has the statistical analysis been performed appropriately and rigorously? 

Reviewer #1: Yes

Reviewer #2: N/A

3. Have the authors made all data underlying the findings in their manuscript fully available?

Reviewer #1: Yes

Reviewer #2: Yes

4. Is the manuscript presented in an intelligible fashion and written in standard English?

Reviewer #1: Yes

Reviewer #2: Yes

5. Review Comments to the Author

Reviewer #1: Professional soccer practitioners’ perceptions of using performance analysis

technology to monitor technical and tactical player characteristics within an academy environment

Thank you for the opportunity to review this relevant manuscript. I welcome studies that introduce the professional soccer practitioners’ perceptions of the application of performance analysis technology within an academy environment and try to elucidate the perceived importance that professional soccer club practitioners place on monitoring technical and tactical player characteristics, current performance analysis practices, and barriers to implementing wearable technology. In fact, I am open to be persuaded to deep understand this relevant topic.

Hence, I am some sympathy with the author's intentions. In addition, the authors provide a decent description of the introduction, process, and some proposals for future research. The topic represents contemporary interest, and the scope of the work and is appropriate for Plos One. I think it is very important to conduct studies like this one, because provide different key points to bring to light studies whose aim is to review what has been published previously on this topic in a rigorous way. This study contribute to the understanding the importance that professional soccer club practitioners place on monitoring technical and tactical player characteristics and the barriers to implementing wearable technology to aid with monitoring within an academy setting. In fact, this study is particularly important because, as the authors indicate, the publications on this topic have a special gap, so it is necessary to group all these papers to have a better understanding of the results obtained previously.

From my point of view, the main strength of the manuscript is the methodology and the results presented clearly (qualitative and quantitative). Apart from this general commentary about the review, more details of some parts of the manuscript (strengths, weakness and questions) are found hereafter.

Abstract.

1. For future study should be indicated at the end of the abstract as well as in the conclusions section some directions about the results in relation to the work

Introduction

2. It is well written and structured. It is a good starting point to place the reader. However, truly think that is very short and need more information and important literature.

3. The introduction doesn’t explain the basic concepts used in the paper neither qualitative nor quantitative similar analysis. Some of these are presented in the methodology section, in terms of how they were measured, however it would have been necessary to clarify their general content in the introduction section.

4. The last part of the introduction (5º last paragraphs since “Implementing technology ….”) are confused and hard to understand.

Methodology

5. Part A. To rewrite the section participants.

6. Table 1. Add information about the variables (measures of each value) methodology clearly explained and justified.

7. Is necessary describe the sample more deeply. Exclusion and inclusion criteria

8. Please, add information about the sample size calculation with G-Power or another software.

9. A schematic representation is necessary to elucidate the study.

10. Part B. Add the same information solicited in Part A

11. Statistical analysis. (What software was used to Part A and for Part B?)

12. Add information about values of effect sizes of Pearson’s chi-squared

Results.

13. I suggest the possibility to create a logistic regression, because is used to predict the outcome of a categorical variable based on the independent or predictor variables. Is possible to create in part A?

14. Please add information about effect size, maybe V-Cramer or othr. In fact, when you have information about effect size you can discuss more precise the discussion.

15. Truly, I think that the figure 1 and figure 2 is not necessary if the information is added in the text. The same with figure 4 and 5.

16. In Part-B interviews. I need that you elucidate the information about categories and their categorizations and also what coding of data collections was used? I understand that you use software similar to WEFT QDA, NVIVO, QUIRKOS?

17. I think that you need reduce the sentences of each question.

18.

Discussion

19. Discussions is appropriate, making reference to the results of other studies, but, the theoretical and practical implications of the research are vaguely mentioned. It necessary improve it.

Conclusions

20. The conclusions respond to the objectives of the study but prospective should be included.

Reviewer #2: The introduction provides a good overview of the topic. It relies on up-to-date literature, and it is written fashionably.

One question is regarding the lack of literature addressed by the study. I see value in the research problem addressed, but the authors could present deeply previous research on the topic – for example, previous studies that highlighted coaches’ and analysts’ perceptions about training and analysis practices within soccer (or maybe other team sports). This would allow a better comprehension of the current study to the state-of-the-art.

Also, the rationale presented in the introduction relates to the TID. However, the aim is written more openly, so it is unclear whether the authors are aiming at investiganting the role of technology in TID or in soccer. If it is not just in the TID, it would be required to improve the introduction.

The abovementioned issue is relevant as many practitioners selected for the interviews are not directly involved in the talent identification process (maybe more related to talent development when the player is already in the club).

Methods

It is unclear why one single academy/club was selected. This selection might bias the result as the practitioners’ perceptions are all influenced by the same team’s principles and methodologies. Therefore, the authors cannot investigate the “professional soccer practitioners’ perceptions of the application of performance analysis technology” (L101) but investigate how this specific team deals with this question instead.

Results

I’d like to recommend that the authors seek a more easy-to-read data visualization strategy as the current tables are large and hard to follow.

Do all the practitioners know the difference between technical metrics and tactical metrics? In some cases, this difference is subtle (or even nonexistent), and I wonder whether professionals from different areas can adequately differentiate them before answering the questionnaire.

Discussion

Many of the cited studies were first introduced in the discussion, while some would help the authors better justify the study in the introduction.

6. PLOS authors have the option to publish the peer review history of their article (what does this mean?). If published, this will include your full peer review and any attached files.

Reviewer #1: No

Reviewer #2: No

---

## [Author Response · Author response to Decision Letter 0]

31 Dec 2023

These guidelines have now been followed and the manuscript updated accordingly 

Informed consent was provided. This was written for surveys and verbal (audio recorded) for interviews. No minors were involved. 

No medical data was used. 

Thank you. We do not wish to make changes to this and will provide access on acceptance

---

## [Decision Letter · Decision Letter 1]

23 Jan 2024

Professional soccer practitioners’ perceptions of using performance analysis technology to monitor technical and tactical player characteristics within an academy environment: A category 1 club case study

PONE-D-23-34955R1

Dear Dr. Davidson,

We’re pleased to inform you that your manuscript has been judged scientifically suitable for publication and will be formally accepted for publication once it meets all outstanding technical requirements.

Kind regards,

Filipe Manuel Clemente, PhD

Academic Editor

PLOS ONE

Additional Editor Comments (optional):

Reviewers' comments:

Reviewer's Responses to Questions

**Comments to the Author**

1. If the authors have adequately addressed your comments raised in a previous round of review and you feel that this manuscript is now acceptable for publication, you may indicate that here to bypass the “Comments to the Author” section, enter your conflict of interest statement in the “Confidential to Editor” section, and submit your "Accept" recommendation.

Reviewer #1: All comments have been addressed

Reviewer #2: All comments have been addressed

2. Is the manuscript technically sound, and do the data support the conclusions?

Reviewer #1: Yes

Reviewer #2: Yes

3. Has the statistical analysis been performed appropriately and rigorously? 

Reviewer #1: Yes

Reviewer #2: N/A

4. Have the authors made all data underlying the findings in their manuscript fully available?

Reviewer #1: Yes

Reviewer #2: No

5. Is the manuscript presented in an intelligible fashion and written in standard English?

Reviewer #1: Yes

Reviewer #2: Yes

6. Review Comments to the Author

Reviewer #1: (No Response)

Reviewer #2: I believe all major issues were addressed. From my perspective, the article is suitable for publication.

7. PLOS authors have the option to publish the peer review history of their article (what does this mean?). If published, this will include your full peer review and any attached files.

Reviewer #1: No

Reviewer #2: No

---

## [Editor Report · Acceptance letter]

28 Feb 2024

PONE-D-23-34955R1 

PLOS ONE

Dear Dr. Davidson, 

I'm pleased to inform you that your manuscript has been deemed suitable for publication in PLOS ONE. Congratulations! Your manuscript is now being handed over to our production team.

Kind regards, 

on behalf of

Dr. Filipe Manuel Clemente 

Academic Editor

PLOS ONE